# Cryopreservation of Human Adipose Tissues and Adipose-Derived Stem Cells with DMSO and/or Trehalose: A Systematic Review

**DOI:** 10.3390/cells10071837

**Published:** 2021-07-20

**Authors:** Conor A. Crowley, William P. W. Smith, K. T. Matthew Seah, Soo-Keat Lim, Wasim S. Khan

**Affiliations:** 1Australasian College of Cosmetic Surgery, Parramatta, NSW 2150, Australia; crowleca@tcd.ie; 2School of Clinical Medicine, University of Cambridge, Cambridge CB2 0SP, UK; wpws3@cam.ac.uk; 3Division of Trauma and Orthopaedic Surgery, Addenbrooke’s Hospital, University of Cambridge, Cambridge CB2 0QQ, UK; wk280@cam.ac.uk; 4The Ashbrook Cosmetic Surgery, Mosman, NSW 2088, Australia; sookeatlim@gmail.com

**Keywords:** adipose-derived stem cells, cryopreservation, trehalose, lipoaspirate, autologous fat grafting, ageing

## Abstract

Adipose tissue senescence is implicated as a major player in obesity- and ageing-related disorders. There is a growing body of research studying relevant mechanisms in age-related diseases, as well as the use of adipose-derived stem cells in regenerative medicine. The cell banking of tissue by utilising cryopreservation would allow for much greater flexibility of use. Dimethyl sulfoxide (DMSO) is the most commonly used cryopreservative agent but is toxic to cells. Trehalose is a sugar synthesised by lower organisms to withstand extreme cold and drought that has been trialled as a cryopreservative agent. To examine the efficacy of trehalose in the cryopreservation of human adipose tissue, we conducted a systematic review of studies that used trehalose for the cryopreservation of human adipose tissues and adipose-derived stem cells. Thirteen articles, including fourteen studies, were included in the final review. All seven studies that examined DMSO and trehalose showed that they could be combined effectively to cryopreserve adipocytes. Although studies that compared nonpermeable trehalose with DMSO found trehalose to be inferior, studies that devised methods to deliver nonpermeable trehalose into the cell found it comparable to DMSO. Trehalose is only comparable to DMSO when methods are devised to introduce it into the cell. There is some evidence to support using trehalose instead of using no cryopreservative agent.

## 1. Introduction

Ageing results in adipose tissue dysfunction, resulting in systemic effects such as peripheral insulin resistance and inflammation. Cellular senescence and progenitor cell dysfunction are also seen in ageing adipose tissues, and these may represent potential therapeutic targets in age-related disease. Adipose tissue comprises many cell types, generally divided into two fractions: the adipocyte fraction (AF), which contains primarily mature adipocytes, and the stromovascular fraction (SVF), which comprises progenitor cells (including adipose-derived stem cells, ADCSs), pericytes, and fibroblasts. Keeping these cells in long-term culture for clinical use or research has inherent limitations, and cell banking of tissue by utilising cryopreservation would allow for much greater flexibility of use [1].

ADSCs also show great promise in regenerative medicine as these cells can differentiate into several different cell types, which can be used for tissue regeneration. For example, ADSCs can differentiate into a large number of cell types, which include chondrogenic, myogenic, osteogenic, angiogenic, and neuronal lineages [2]. With regard to the identification of ADSCs, the Mesenchymal and Tissue Stem Cell Committee of the International Society for Cellular Therapy have proposed minimal criteria to define human mesenchymal stem cells. First, they must be plastic-adherent. Second, they must express the cell surface markers CD105, CD73 and CD90, and lack the expression of CD45, CD34, CD14 or CD11b, CD79alpha or CD19 and HLA-DR. Third, they must differentiate to osteoblasts, adipocytes and chondroblasts in vitro [3].

Various methods of stromal vascular fraction cryopreservation have been demonstrated [4], but the protocol which maximises cell viability remains unclear. Biospecimens have to reach cryogenic temperatures (−196 °C) before long-term storage in liquid nitrogen. They are then warmed back to normothermic temperature (37 °C) as necessary. Two methods have been described for cryopreservation: slow-freezing and vitrification. Slow-freezing cools specimens at a low rate of about 1 °C/min to reduce intracellular ice formation. The disadvantage of this technique is that it can cause excessive dehydration, leading to osmotic stress, and deformation. Vitrification involves rapid cooling without ice formation but requires a high concentration of cryoprotective agent (CPA), and there may be limitations on the sample volume [5]. Both methods require adding a CPA to adipose tissue prior to cooling, which is a critical step in the optimal cryopreservation of human ADSCs.

Dimethyl sulfoxide (DMSO) is the most commonly used CPA in the cryopreservation of ADSCs. DMSO is a small amphipathic molecule with two non-polar methyl groups and a polar sulfoxide group. The solvent activity of DMSO allows it to readily cross cell membranes. DMSO helps prevent the formation of intracellular ice, thereby protecting the tissue from freezing injuries and helping achieve maximum tissue survival during the freezing and thawing process. DMSO is toxic to cells at room temperature and washing and centrifugation are needed after thawing to remove it from the tissue [6]. This is particularly important in clinical settings when stem cells are administered in vivo as a cell therapy, where inadvertent administration of DMSO can cause severe side effects or death [7]. Furthermore, high centrifugation speeds during the washing and removal of CPAs may damage the fragile adipose tissue after freezing/thawing [8]. The ideal cryopreservation agent should be non-toxic for cells and patients, non-antigenic and chemically inert. It should ideally not require washing prior to transplantation [9].

Owing to their low toxicity, sugars have been used as extracellular cryopreservation agents. Trehalose (α-d-glucopyranosyl α-d-glucopyranoside) is a disaccharide of glucose with low toxicity in contrast to DMSO. It is synthesised by species of bacteria, fungi, yeast, insects and plants that are prone to dehydration but is absent from vertebrates [10]. Such organisms can withstand dehydration and extreme cold. The suggested mechanisms for the protein-stabilising effect of trehalose include water replacement, glass transition and chemical stability [11]. Trehalose acts to replace water molecules and form hydrogen bonds with proteins, maintaining protein conformation in conditions of dehydration [12]. In the dehydrated state, trehalose forms a non-crystalline glass that reduces molecular movement, inhibiting metabolic activity and protein breakdown. Trehalose also acts to stabilise the phospholipid bilayer when in its “clam-shaped” conformation [13]. As the plasma membrane of mammalian cells is impermeable to trehalose, intracellular delivery is vital for cryopreservation [14,15]. The precise mechanism by which trehalose provides such protection is yet to be fully elucidated and recent studies have demonstrated adequate outcomes with only trehalose in the cryopreservation of various tissues [16,17,18,19,20]. Autologous fat could potentially be banked and used later without having to remove any agent from the cryopreserved fat grafts while providing useful long-term cryopreservation [21,22,23]. This result may be through a possible synergistic mechanism.

In an update on the cryopreservation of adipose tissue and ADSCs, Shu et al. [24] highlighted that slow-freezing (1–2 °C/min) with a combination of CPAs and rapid thawing produced optimal results and that ADSCs can retain high degrees of cell viability. This is the first review that specifically examines the role of trehalose as a CPA in the cryopreservation of human adipose tissue and ADSCs.

## 2. Materials and Methods

A systematic review was performed using the following databases: PubMed, CINAHL, Web of Science, MEDLINE, Cochrane Library and EMBASE. We included the following keywords and search terms: adipose stem cells, cryopreservation, trehalose, lipoaspirate, autologous fat grafting and cryopreservation. Those keywords were searched in that order in each database. We used a snowballing technique with an arbitrary cut-off of greater than two hundred articles for each database. An initial reviewer performed the search before being checked by a second reviewer. Any disputes about inclusion were resolved by a third reviewer.

The following inclusion criteria were applied: (1) interventional studies using trehalose in the cryopreservation of human adipose-derived stem cells, including case series, case–control, cohort studies and randomised controlled trials; (2) studies translated into the English language; and (3) full article accessible.

Exclusion criteria: (1) unpublished literature; (2) literature published before 2000; (3) articles not available for free viewing; and (4) studies using animal adipose-derived stem cells.

A total of 430 articles were retrieved. A total of thirteen articles with fourteen suitable studies were relevant for this systematic literature review. The studies included and excluded are summarised in Figure 1. Formal statistical analysis was not performed because of the methodologic heterogeneity demonstrated among the articles.

## 3. Results

### 3.1. Combined Use of Trehalose and DMSO for Cryopreservation

Eight studies were included and are described in Table 1. All studies report improved viability of adipocytes following a freeze/thaw cycle if a CPA (trehalose and DMSO combined) was used compared to controls where no CPA was used. A variety of concentrations of trehalose and DMSO were used, and no significant conclusions can be drawn about the optimum concentration/combinations for optimum adipocyte yield. The cells were washed to remove the CPA according to standard protocols.

Pu et al. [25] tested their hypothesis that adipose aspirates obtained from conventional lipoplasty could be cryopreserved. They cryopreserved adipose tissues using slow cooling and fast rewarming rates. The combination of trehalose and DMSO resulted in significantly more viable adipocytes with better function than simple cryopreservation with liquid nitrogen. They concluded that the combination appears to provide adequate preservation of adipose tissues obtained from conventional lipoplasty.

In a later study, Pu et al. [25] injected adipose aspirates into the posterior scalp of athymic nude mice. They assessed the gross appearance of the fat grafts for volume, weight and histology. The trehalose and DMSO combinations maintained better weight, volume and tissue architecture than the simple cryopreservation group. The results of both cryopreserved groups were less satisfactory than the fresh fat control group. They concluded that the DMSO and trehalose combination provided reasonably good cryopreservation of adipose tissues. De Rosa et al. [26] tested different concentrations of CPA. When trehalose and DMSO were combined, cells recovered more than 80% of their viability post cryopreservation, with antigen expression levels close to those seen in freshly isolated cells.

Cui et al. [27] compared trehalose alone to a combination of DMSO and trehalose. They injected cryopreserved fat grafts into the posterior scalp of a nude mouse. They found that trehalose was comparable to DMSO and trehalose in providing adequate protection during cryopreservation.

Pu et al. [22] found in an in vitro study that cryopreservation with trehalose and DMSO combined yielded on average 90% of adipocytes compared to the fresh fat control. Cui et al. [23] investigated different concentrations of CPAs in vitro and found that 0.5 mol/L DMSO and 0.2 mol/L trehalose were the most effective and were significantly better than simple cryopreservation without CPAs. They went on to perform an in vivo study using 0.5 mol/L DMSO and 0.2 mol/L trehalose to cryopreserve adipose aspirates. The adipose aspirates were injected into a nude mouse and compared to controls. The DMSO and trehalose group had significantly more retained weight and volume of the injected fat grafts than the simple cryopreservation group. Pu et al. [28] tested a combination of 0.5 mol/L DMSO and 0.2 mol/L trehalose. They found that the combination of CPAs maintained a similar number of viable adipocytes and histology of the cells to the fresh fat control. However, they did report that glycerol-3-phosphatase dehydrogenase activity was significantly lower than the control, indicating a decrease in adipocyte cellular function in this group.

### 3.2. Comparing Trehalose vs. DMSO for Cryopreservation

Four Level 3 studies were available for analysis (Table 2). Two studies showed noninferior outcomes when trehalose was used as a CPA [7,15] when compared with DMSO, while two studies showed inferior outcomes [29,30]. Inferior outcomes were seen when no additional techniques were used to encourage the intracellular trafficking of trehalose (e.g., electroporation) [29,30].

Rao et al. [15] examined trehalose as a sole cryoprotectant. They achieved effective delivery into human ADSCs (hADSCs) and reported that preincubation with trehalose significantly increased post-cryopreservation cell viability to 88.1% ± 7.5%. The viability is further improved to 91.2% ± 3.4% by using a higher concentration of trehalose. This observed cell viability was similar to the DMSO group (88.2% ± 2.2%), indicating that trehalose was successfully delivered into the hADSCs during cryopreservation. Dovgan et al. [7] demonstrated that electroporation is an efficient method for delivering trehalose into hADSCs. They achieved a high survival rate approaching the values obtained with standard DMSO protocol (91.5% ± 1.6%). The authors observed no statistically significant difference between 250 mmol/L trehalose and DMSO, and with only a slight difference between 400 mmol/L trehalose and DMSO.

Roato et al. [29] compared DMSO and trehalose, and both agents showed similar degrees of viability of samples after thawing. However, the hADSCs cryopreserved in DMSO had a more rapid growth and arrived at confluence in a few days with a better morphology than those treated with trehalose. They concluded that DMSO is superior to trehalose for the cryopreservation of adipose tissue. Yong et al. [30] compared the effects of different combinations of trehalose, DMSO and FBS on hADSCs in terms of cell phenotype, proliferation potential, differentiation potential, stemness and viability. They found that ADSCs preserved in 250 mmol/L trehalose showed the lowest degree of cell viability (*P* < 0.05).

### 3.3. Comparing Trehalose with Fresh Fat Control or Simple Cryopreservation

Four studies were included for analysis (Table 3). Study protocols were heterogenous and included studies that used nanoparticles to mediate the intracellular delivery of trehalose [15], while others did not. The data appear to suggest that the use of a CPA (such as trehalose) has better outcomes than not using a CPA in terms of cell viability and cell differentiation after freezing and thawing.

Cui et al. [27] examined the use of trehalose as a single CPA of adipose aspirates and found that a concentration of 0.35 mol/L appeared to provide optimal cryopreservation. They also found that when a glycerol-3-phosphate dehydrogenase assay was performed to assess cellular function, there was no difference between any of the trehalose groups and the fresh fat control. In the second part of the study, cryopreserved fat was injected into the posterior scalp of a nude mouse. They found that at 4 months the volume and weight of the grafts were significantly lower than those of the fresh fat controls (*P* < 0.5). Rao et al. [15] used nanoparticle-mediated intracellular delivery of trehalose into adipocytes and found that the morphology and differentiation of cells were comparable between the fresh fat control group and the trehalose group. Pu et al. [31] and Cui et al. [23] reported that the trehalose group had a significant reduction in viable adipocytes compared to the fresh fat control. Both authors also said that the trehalose groups had significantly higher viable adipocyte counts than the simple cryopreservation group that used no CPA. It is important to note that cryopreservation is not typically carried out without a CPA so there is limited utility in this comparison. 

## 4. Discussion

There are several limitations to this review. Firstly, there is an English language bias in our search strategy as only articles published in English were reviewed. However, a study by Juni et al. [32] found that this has little effect on the estimation of summary treatment effect. Secondly, several variables influence the viability of cryopreserved adipose tissue, including harvest technique, storage temperature and duration, thawing rate and the associated surgical procedure [24]. The individual contribution of each factor is not clearly understood, therefore making it difficult to draw comparisons between the studies. Thirdly, tissues were frozen for short times in several studies, which is not typical of tissue and cell banking applications.

The results from eight studies in our systematic review showed that DMSO and trehalose could be combined effectively to cryopreserve adipocytes. The advantage of this combination is that it reduces the dose of DMSO required. All four studies comparing trehalose and DMSO to simple cryotherapy with liquid nitrogen alone found a significant improvement in cryopreservation [21,23,25]. Pu et al. [6,22,25,28] found in four studies that the combination of trehalose and DMSO acted as successful CPAs with the number of adipocytes comparing very well to the fresh fat controls. In contrast, Cui et al. [23,27] found that this combination of CPAs was inferior to the fresh fat control groups. Interestingly De Rosa et al. [26] found that the combination of trehalose and DMSO performed better than DMSO alone. This result is in line with the hypothesis that these two CPAs may have a synergistic effect.

To maximise cell viability, a balance is required between reducing intracellular ice formation (by cooling slowly) and cooling quickly enough to minimise the solution effects. Mazur proposed a two-factor hypothesis attributing a lower cell survival rate when the cooling rate is slower than optimal (due to alterations in the solute concentrations of the extracellular and intracellular solutions). A faster cooling rate than optimal also decreases cell survival due to intracellular ice formation [33,34,35,36]. The storage temperature is an important variable. Erdim et al. [37] found that the storage of adipose tissue at 4 °C reduced the number of live fat cells compared to freshly isolated cells, but this was not statistically significant (*P* > 0.05). Wolter et al. [38] compared storing cryopreserved fat cells at −20 °C and −80 °C for up to 30 days. They found that −80 °C yielded higher viability than −20 °C, with the latter leading to nonviable tissue.

Storage temperature is thought to significantly influence the viability of cryopreserved adipose tissue. Subzero temperatures result in ice formation damage, but this can be mitigated by the storage of tissue at less than −70 °C. Rapid thawing can improve cell viability by reducing the risk of intracellular ice formation, and subsequent ice crystal growth. Son et al. [39] performed a study that showed that adipocytes’ viability rapidly declined following frozen storage for one day at both −15 °C and −70 °C and subsequently decreased slowly in storage after eight weeks. By this point, only around 5% of the fat cells remained viable, suggesting that both these storage conditions are not ideal for cryopreservation. Hwang et al. [40] compared the survival rates of adipose tissue in various thawing conditions, such as natural thawing at 25 °C for 15 min; natural thawing at 25 °C for 5 min, followed by rapid thawing at 37 °C in a water bath for 5 min; and rapid thawing at 37 °C for 10 min in a water bath. The authors found a rapid thaw at 37 °C for 10 min in a water bath to be the optimal method for thawing the cryopreserved adipose tissue. The freezing behaviour of the cells can be modified by the addition of CPAs, which affect the rates of water transport, ice nucleation and ice crystal growth. MacRae et al. [41] cryopreserved adipose tissue with and without CPAs. They concluded that at −196 °C, adipose tissue preserved with CPAs showed increased viability compared to that preserved without CPAs, thus demonstrating their importance.

Long-term cryopreserved ADSCs demonstrate a low risk of tumorigenicity and therefore offer great potential in regenerative medicine [30]. Developing successful serum-free and xeno-free defined cryopreservation methods will enable more widespread clinical applications of stem cells [42]. The literature suggests a combination of extracellular trehalose with a penetrating CPA appears to be efficient [42], and the potential benefits of using trehalose as a CPA are well-described. However, delivering sufficient quantities intracellularly remains a hurdle. Various techniques have been described, including microinjection into the cell, electroporation, fluid phase endocytosis, engineered transmembrane pores and nanoparticles. The optimum method for this is still under investigation. Outcomes in the literature are heterogenous because cryopreservation protocols may result in insufficient trafficking of trehalose into cells, or the delivery step (e.g., electroporation) might compromise cellular integrity and affect their ability to withstand further stresses induced by the freezing [43]. Further studies are therefore required to develop an effective approach for the intracellular delivery of trehalose for cell cryopreservation.

Finally, there remains some variation in the cryopreservation of tissues and further studies are required to determine optimal protocols. Choudhery et al. demonstrated that adipose tissue could be successfully cryopreserved without compromising cell morphology, as well as subsequent proliferation and differentiation potential [1], making this a useful tool in regenerative medicine. However, more work needs to be performed on characterising the optimal cryopreservation techniques for the different tissues and cell types of interest.

## Figures and Tables

**Figure 1 cells-10-01837-f001:**
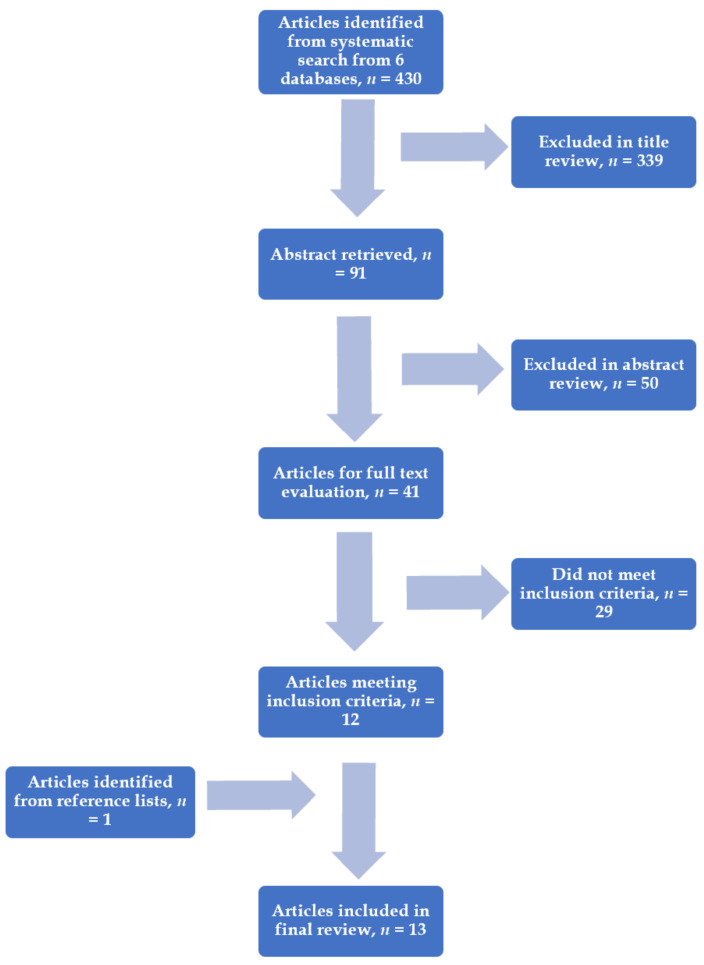
Flow diagram showing the literature search.

**Table 1 cells-10-01837-t001:** Cryopreservation with trehalose and dimethyl sulfoxide combined.

Ref.	Level of Evidence OCEBM	Methods, Slow Cooling and Rapid Rewarming	CPA Concentration mol/L, Groups	Temperature, °C	Time	Results
Pu et al. [21] 2004	Level 3	Adipose aspirates from lipoplasty were randomised to each group and were evaluated by viable adipocyte counts, G3PDH assay and routine histology.	(1) Simple cryopreservation with liquid nitrogen only; (2) 0.5 mol/L DMSO + 0.2 mol/L trehalose; and (3) Control group with fresh adipose	−196	20 min	Group 2 showed significantly higher adipocyte viability and superior cellular function of adipocytes compared to group 1. (2.15 ± 0.68 vs. 1.04 ± 0.35 × 10^6^/mL, *P* < 0.0001)
Pu et al. [25] 2006	Level 3	Cryopreserved adipocytes from cosmetic lipoplasty were administered into the posterior scalp of a nude mouse. Gross appearance of fat grafts was observed for up to 16 weeks. At the end of the study, final graft weight and volume and corresponding histology were determined.	(1) 0.5 mol/L DMSO + 0.2 mol/L trehalose solution; (2) Simple cryopreservation with liquid nitrogen only; and (3) Control group with fresh adipose aspirates	−196	20 min	Group 1 showed greater graft weight, volume and retained tissue architecture when compared to group 2 (*P* < 0.0001); the fresh control group showed a greater retained volume (47.7% ± 18.6%), and this was statistically significant relative to both groups 1 (35.3% ± 7.8%, *P* < 0.05) and 2 (6.5% ± 3.7%, *P* < 0.0001).
De Rosa et al. [26] 2009	Level 3	Different concentrations of CPA were evaluated to preserve ADSCs for future clinical applications.	(1) 1% DMSO (0.1 mol/L), 9% trehalose (0.26 mol/L), 90% FBS; (2) 4% DMSO (0.6 mol/L), 6% trehalose (0.18 mol/L), 90% FBS; (3) 8% DMSO (1.1 mol/L), 2% trehalose (0.06 mol/L), 90% FBS; and (4) 10% DMSO (1.4 mol/L), 90% FBS	−196	1, 6 and 12 months	The best freezing solution consisted of 90% FBS, 4% DMSO (0.6 mol/L) and 6% trehalose. The thawed cells in this group showed superior differentiation efficiency and higher levels of antigen expression, similar levels found in fresh isolates.
Cui et al. [27] 2010	Level 3	0.5 mL of cryopreserved adipocyte grafts was thawed and injected into the posterior scalps of mice for 8 weeks. Graft volume, weight and histology were evaluated at the end of the study.	(1) 0.5 mol/L DMSO + 0.2 mol/L trehalose; (2) 0.35 mol/L trehalose; and (3) Control (fresh fat graft)	−196	20 min	Groups 1 and 2 showed no statistically significant difference in maintained volume (vs. 46.1% ± 14.4% vs. 38.2% ± 10.1%, NS) and weight (38.9% ± 14.7% vs. 34.1% ± 12.1%*,* NS). Both cryopreservation groups were found to be inferior to control (both *P* < 0.05).
Pu et al. [22] 2006	Level 3	In vitro study measuring the rate of growth and viable cell count (after 2 weeks) of fresh vs. cryopreserved (with fast rewarming) adipocyte aspirates.	(1) 0.5 mol/L DMSO + 0.2 mol/L trehalose and (2) Control fresh adipose aspirates	−196	20 min	The cryopreserved aspirates produced 90% of the cell count from fresh aspirates (3.7 ± 1.4 × 10^5^ processed lipoaspirate cells per millilitre aspirates vs. 4.1 ± 1.4 × 10^5^ cells/mL).
Cui et al. [23] 2007	Level 3	Different CPAs and their concentrations were tested in vitro.	(1) Fresh adipose aspirates; (2) Cryopreserved adipose aspirates without cryoprotectants; and (3) Cryopreserved adipose aspirates with cryoprotectants—0.2 mol/L DMSO + 0.1 mol/L trehalose, 0.5 mol/L DMSO + 0.2 mol/L trehalose, 0.25 mol/L trehalose, 0.5 mol/L trehalose, 1.0 mol/L DMSO, 1.5 mol/L DMSO	−196	20 min	The combination of 0.5 mol/L DMSO and 0.2 mol/L trehalose produced the greatest adipocyte count; group 3 produced a significantly higher adipocyte count than group 2 (2.06 ± 0.54 × 10^6^/mL vs. 1.07 ± 0.41 × 10^6^/mL, *P* < 0.0011); group 1 displayed only a marginally higher adipocyte count relative to group 3 (vs. 2.57 ± 0.56 × 10^6^/mL vs. 2.06 ± 0.54 × 10^6^/mL, *P* = 0.083); group 3 displayed less tissue shrinkage relative to group 2.
Cui et al. [23] 2007	Level 3	1 mL fat graft was injected into nude mice and subsequently harvested 4 months later and analysed for volume, weight and histology. Maintenance of tissue architecture was rated as per the following scale: “5-pristine cellular architecture in all sections examined; 4-mild disruption of cellular architecture in <50% of sections; 3-mild disruption of cellular architecture in >50% of sections; 2-severe disruption of cellular architecture in <50% of sections; 1-severe disruption of cellular architecture in >50% of sections”.	(1) Fresh adipose aspirates; (2) Cryopreserved adipose aspirates without CPAs; and (3) Cryopreserved adipose aspirates with CPAs—0.5 mol/L DMSO + 0.2 mol/L trehalose	−196	1 week	Retained graft volume and weight were significantly higher in group 3 compared to group 2 (both *P* < 0.0001); histology showed extensive tissue fibrosis in group 2, in contrast to relatively preserved tissue architecture with very little fibrosis in group 3; the mean histological rating score in group 1 was significantly higher than that of group 2 (4.60 ± 0.22 vs. 1.50 ± 0.26, *P* < 0.0001).
Pu et al. [28] 2010	Level 3	The fat graft samples from both groups were evaluated with trypan blue vital staining, G3PDH assay and routine histology.	(1) 0.5 mol/L DMSO and 0.2 mol/L trehalose and (2) Fresh fat graft control	−196	20 min	Groups 1 and 2 showed similar adipocyte counts (3.46 ± 0.91 vs. 4.12 ± 1.11 × 10^6^/mL, *P =* 0.22); activity of G3PDH was significantly higher in group 2 compared with group 1 (0.66 ± 0.09 vs. 0.47 ± 0.09 U/mL, *P* < 0.001); histological analysis showed mainly normal structure of fragmented fatty tissues in both groups.

OCEBM: Oxford Centre for Evidence-Based Medicine; CPA: Cryoprotective agent; G3PDH: Glycerol-3-phosphate dehydrogenase; DMSO: Dimethyl sulfoxide; FBS: Foetal bovine serum; NS: Not significant; ADSC: Adipose-derived stem cell.

**Table 2 cells-10-01837-t002:** Cryopreservation with trehalose vs. dimethyl sulfoxide.

Ref.	Level of Evidence OCEBM	Methods, Slow Cooling and Rapid Rewarming	CPA Concentration mol/L, Groups	Temperature, °C	Time	Results
Rao et al. [15] 2015	Level 3	To determine the cryopreservation of primary hADSCs using nanoparticle-mediated intracellular delivery of trehalose as the sole cryoprotectant.	(1) 0.2 mol/L trehalose; (2) 100 mL/L DMSO; and (3) Fresh control	−196	1 day	Trehalose acted as a successful CPA; cryopreservation with trehalose resulted in similar cell survival as compared to DMSO.
Dovgan et al. [7] 2016	Level 3	The efficiency of combining reversible electroporation and trehalose for cryopreservation of hADSCs.	(1) DMSO; (2) 0.25 mol/L trehalose with electroporation; and (3) 0.4 mol/L trehalose without electroporation	−196	1 week	No statistically significant difference between DMSO (91.5% ± 1.6%) and 250 mmol/L trehalose (83.8% ± 1.8%) treated with electroporation was observed, with a slight difference between DMSO and 0.4 trehalose without electroporation (78.4% ± 1.5%).
Roato et al. [29] 2016	Level 3	To evaluate ADSC viability and differentiation capability after cryopreservation.	(1) FBS + 10% DMSO and (2) FBS + 0.35 mol/L trehalose	−196	3 days	DMSO is superior to trehalose for cryopreservation of adipose tissue. Cell cultures demonstrated that ADSCs isolated from lipoaspirates cryopreserved in DMSO showed a higher growth rate and arrived at confluence in a few days with a better tissue architecture, compared to the cells preserved with trehalose.
Yong et al. [30] 2015	Level 3	To compare the effects of various combinations of CPA on hADSCs in terms of cell phenotype, proliferation potential, differentiation potential, stemness and viability.	(1) 0.25 mol/L trehalose; (2) 5% DMSO (0.7 mol/L); (3) 10% DMSO (1.4 mol/L); (4) 5% DMSO (0.7 mol/L) + 20% FBS; (5) 10% DMSO (1.4 mol/L) + 20% FBS; and (6) 10% DMSO (1.4 mol/L) + 90% FBS	−196	3 months	5% DMSO without FBS may be an ideal CPA for efficient long-term cryopreservation of hADSCs. ADSCs preserved in 0.25 mol/L trehalose showed the lowest cell viability (*P* < 0.05).

OCEBM: Oxford Centre for Evidence-Based Medicine; CPA: Cryoprotective agent; hADSCs: Human adipose-derived stem cells; DMSO: Dimethyl sulfoxide; FBS: Foetal bovine serum.

**Table 3 cells-10-01837-t003:** Cryopreservation using trehalose vs. fresh fat control or simple cryopreservation.

Ref.	Level of Evidence OCEBM	Methods, Slow Cooling and Rapid Rewarming	CPA Concentration mol/L, Groups	Temperature, °C	Time	Results
Cui et al. [6] 2009	Level 3	Adipose aspirates were cryopreserved using trehalose as a CPA in seven different concentrations and compared to a fresh fat control group for viability in vitro. A G3PDH assay was also performed to assess intracellular function.	Trehalose: (1) 0.20 mol/L; (2) 0.25 mol/L; (3) 0.30 mol/L; (4) 0.35 mol/L; (5) 0.40 mol/L; (6) 0.50 mol/L; (7) 0.75 mol/L; and (8) Control	−196	20 min	Cryopreservation with 0.35 mol/L trehalose was found to preserve the most adipocytes. This concentration of trehalose showed no statistical difference relative to control (2.4 ± 0.52 vs. 1.88 ± 0.61 × 10^6^/mL; *P* > 0.05). No concentration of trehalose showed a significant difference in intracellular function relative to control (all *P* > 0.05).
Cui et al. [27] 2010	Level 3	0.5 mL of cryopreserved fat grafts was thawed and injected into the posterior scalps of mice for 8 weeks. Weight, volume and histology of grafts were analysed at the end of the study.	(1) 0.5 mol/L DMSO + 0.2 mol/L trehalose; (2) 0.35 mol/L trehalose; and (3) Control (fresh fat graft)	−196	20 min	Group 2 and group 1 were inferior to the control group (both *P* < 0.05). There was a significantly higher percentage of maintained volume of injected fat in the control group (55.5% ± 11.7%) compared to group 1 (46.1% ± 14.4%, *P* < 0.05) or group 2 (38.2% ± 10.5%, *P* < 0.01). The control group showed a significantly higher maintained weight relative to both groups 1 (38.9% ± 14.7%, *P* < 0.01) and 2 (34.1% ± 12.1%, *P* < 0.01).
Rao et al. [15] 2015	Level 3	To determine the cryopreservation of primary hADSCs using nanoparticle-mediated intracellular delivery of trehalose as the sole CPA.	(1) 0.2 mol/L trehalose; (2) 100 mL/L DMSO; and (3) Fresh control	−196	1 day	hADSCs’ tissue architecture post cryopreservation with trehalose is similar to that of fresh isolates. Trehalose maintained comparable differentiation capabilities of the cryopreserved vs. fresh hADSCs. Trehalose acted as a successful cryoprotectant.
Pu et al. [31] 2005	Level 3	The efficacy of trehalose as the sole CPA for cryopreservation of adipocytes, with the aim to develop a protocol which enables optimal preservation of adipose tissues.	(1) Control fresh adipose aspirates; (2) Simple cryopreservation group: cryopreserved adipose aspirates without CPAs; and (3) Optimal cryopreservation group: 0.25 mol/L trehalose	−196	20 min	Adipocyte count was significantly higher in group 3 than group 2 (1.78 ± 0.33 vs. 0.99 ± 0.35 × 10^6^/mL, *P* < 0.0001). Adipocyte count in group 3 was significantly lower than fresh isolates (1.78 ± 0.33 vs. 2.64 ± 0.54 × 10^6^/mL, *P* < 0.001). The G3PDH activity in group 3 was also significantly lower than control (0.24 ± 0.07 vs. 0.32 ± 0.09 U/mL, *P* < 0.05). There was a statistically significant increase in G3PDH activity, which was significantly higher in group 3 relative to group 2 (0.24 ± 0.07 vs. 0.15 ± 0.06 U/mL, *P* < 0.01).
Cui et al. [23] 2007	Level 3	In vitro study where different cryoprotectant agents and their concentrations were tested.	(1) Fresh adipose aspirates; (2) Cryopreserved adipose aspirates without CPAs; and (3) Cryopreserved adipose aspirates with CPAs. (1) 0.2 mol/L DMSO + 0.1 mol/L trehalose; (2) 0.5 mol/L DMSO + 0.2 mol/L trehalose; (3) 0.25 mol/L trehalose; (4) 0.5 mol/L trehalose; (5) 1.0 mol/L DMSO; and (6) 1.5 mol/L DMSO	−196	20 min	The viable adipocyte count in the fresh fat control group was still significantly higher than the count in any of the six different CPA groups (all *P* < 0.0001). Significantly higher integrated viable adipocyte count of adipose aspirates was found in group 3 compared with group 2 (2.06 ± 0.54 × 10^6^/mL vs. 1.07 ± 0.41 × 10^6^/mL, *P* < 0.001).

OCEBM: Oxford Centre for Evidence-Based Medicine; CPA: Cryoprotective agent; hADSCs: Human adipose-derived stem cells; DMSO: Dimethyl sulfoxide; G3PDH: Glycerol-3-phosphate dehydrogenase.

## Data Availability

Not applicable.

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
