# Peer review of "Cryopreservation of Human Adipose Tissues and Adipose-Derived Stem Cells with DMSO and/or Trehalose: A Systematic Review"

_cells, 2021, doi:10.3390/cells10071837_

Round 1
Reviewer 1 Report
In the manuscript “Cryopreservation of Human Adipose Tissues and Adipose-Derived Stem Cells with Trehalose: A Systematic Review” the authors summarize studies about the usage of trehalose for the cryopreservation of adipose cells and tissues. The topic is of great and increasing interest for regenerative medicine. The data is well summarized. Some minor concerns need to be addressed.
Minor concerns:
Some references seem to be missing in the introduction. The first reference in the text is “[2]”, the second reference is “[5]”. From then on, the numbering seems okay, but please revise throughout the manuscript anyway. Additionally, the spacing before the reference bracket is inconsistent.
In line 93 and at the end of the second paragraph on page 12 (the line numbering ends after page 4 when the page direction changes from portrait to landscape) the abbreviation “AFT” is used. “Adipose fat tissue”? “Adipose frozen tissue”? Please elaborate.
In lines 93-95, the authors state: “AFT could potentially be banked and used later without having to remove any agent from the cryopreserved fat grafts. Permeable DMSO and nonpermeable trehalose can provide useful long-term cryopreservation[21-23].” Is the “DMSO” in the latter sentence referencing the statement “without having to remove any agent” from the former sentence? If so, please explain why the DMSO does not need to be removed when in combination with trehalose. If not, please rephrase the sentence to clarify that it still needs to be removed and is not referring to the former sentence.
In table 1 the CPA concentration is indicated in mol/L for all but one reference. It would be easier for the reader to compare the results if the CPA concentrations from the De Rosa et al-study were converted (maybe in brackets additionally to the original indication) to this unit, too. (This also applies for table 2, where the unit mL/L is used additionally)
Tables 1 and 2 would benefit from the information, whether the CPA was removed from the thawed tissue by washing (Alternatively, this could be mentioned in the results text).
Line 130: “Seven studies were included and are described in Table 1.” Aren’t there 8 studies in table 1? (it also says 8 in the first sentence of the discussion)
Table 2, Dovgan et al-row: in the results-column it should read “0.4 trehalose withOUT electroporation”
The authors laudibly list several limitations to their review article. Although the following concern is due to the original articles and not the author’s fault, the authors could maybe add the limitation, that many studies evaluated a very short time in which the tissue was frozen. For tissue and cell banking applications, 20 minutes is an unreasonable short time span. Maybe the exclusion of articles not available for free viewing could be mentioned as a limitation, too.
The discussion, especially the first half, could be structured better. For example, the paragraph starting with “Son et al[39] performed a study…” is not really connected to the discussion. Another example: the paragraph about the limitations of the study is directly after the discussion of the results from the studies in table 1 but before the studies from the other tables are discussed. I don’t see any reason why. Also, a new paragraph should start after “… to draw comparisons between the studies.”. However, the whole discussion should be revised in order to present a better-structured text.
The removing of CPA/washing of thawed tissue should be discussed.
Author Response
REVIEWER 1
In the manuscript “Cryopreservation of Human Adipose Tissues and Adipose-Derived Stem Cells with Trehalose: A Systematic Review” the authors summarize studies about the usage of trehalose for the cryopreservation of adipose cells and tissues. The topic is of great and increasing interest for regenerative medicine. The data is well summarized. Some minor concerns need to be addressed.
Minor concerns: Some references seem to be missing in the introduction. The first reference in the text is “[2]”, the second reference is “[5]”. From then on, the numbering seems okay, but please revise throughout the manuscript anyway. Additionally, the spacing before the reference bracket is inconsistent.
|
Thank you for your comments – we have corrected the references. |
In line 93 and at the end of the second paragraph on page 12 (the line numbering ends after page 4 when the page direction changes from portrait to landscape) the abbreviation “AFT” is used. “Adipose fat tissue”? “Adipose frozen tissue”? Please elaborate.
|
The term AFT has been corrected to ‘autologous fat’ |
In lines 93-95, the authors state: “AFT could potentially be banked and used later without having to remove any agent from the cryopreserved fat grafts. Permeable DMSO and nonpermeable trehalose can provide useful long-term cryopreservation[21-23].” Is the “DMSO” in the latter sentence referencing the statement “without having to remove any agent” from the former sentence? If so, please explain why the DMSO does not need to be removed when in combination with trehalose. If not, please rephrase the sentence to clarify that it still needs to be removed and is not referring to the former sentence.
|
If trehalose alone were used, then this does not need to be removed. The sentence has been changed to ‘Autologous fat could potentially be banked and used later without having to remove any agent from the cryopreserved fat grafts while providing useful long-term cryopreservation’ |
In table 1 the CPA concentration is indicated in mol/L for all but one reference. It would be easier for the reader to compare the results if the CPA concentrations from the De Rosa et al-study were converted (maybe in brackets additionally to the original indication) to this unit, too. (This also applies for table 2, where the unit mL/L is used additionally)
|
The units have been made consistent in both tables. |
Tables 1 and 2 would benefit from the information, whether the CPA was removed from the thawed tissue by washing (Alternatively, this could be mentioned in the results text). |
We have added this in the results section ‘The cells were washed to remove the CPA according to standard protocols.’ |
Line 130: “Seven studies were included and are described in Table 1.” Aren’t there 8 studies in table 1? (it also says 8 in the first sentence of the discussion)
|
This has been corrected. |
Table 2, Dovgan et al-row: in the results-column it should read “0.4 trehalose withOUT electroporation”
|
This has been corrected. |
The authors laudibly list several limitations to their review article. Although the following concern is due to the original articles and not the author’s fault, the authors could maybe add the limitation, that many studies evaluated a very short time in which the tissue was frozen. For tissue and cell banking applications, 20 minutes is an unreasonable short time span. Maybe the exclusion of articles not available for free viewing could be mentioned as a limitation, too.
|
This has been modified – all included articles were accessed. We have also added the line ‘tissues were frozen for short times in several studies, which is not typical of tissue and cell banking applications.’ |
The discussion, especially the first half, could be structured better. For example, the paragraph starting with “Son et al[39] performed a study…” is not really connected to the discussion. Another example: the paragraph about the limitations of the study is directly after the discussion of the results from the studies in table 1 but before the studies from the other tables are discussed. I don’t see any reason why. Also, a new paragraph should start after “… to draw comparisons between the studies.”. However, the whole discussion should be revised in order to present a better-structured text.
The removing of CPA/washing of thawed tissue should be discussed.
|
We have restructured the discussion as suggested.
A reference with regards to the effects of centrifugation during washing on adipocyte viability has also been added to the introduction. ‘Furthermore, high centrifugation speeds during the washing and removal of CPA may damage the fragile adipose tissue after freezing/thawing.’ |
Reviewer 2 Report
The manuscript submitted by the authors is potentially of interest but needs modifications to improve it for publication.
A significant contribution to this topic was published several years ago by Choudhary et al but has not been cited or discussed.
If clinical application of the proposed methods has been used, please note and discuss it.
Freezing and thawing of ADSC is simple and readily done as for any single cell solution. In fact, thousands of clinical applications of such cellular products are done successfully each year.
Authors should note that no PI attempts cyropreservation without some CP added, so the comparisons on p14 are unrealistic.
A comparison of DMSO with internal versus external trehalose should be made.
Discussion of different approaches on cell viability, proliferation and differentiation should be made.
Author Response
REVIEWER 2
The manuscript submitted by the authors is potentially of interest but needs modifications to improve it for publication. A significant contribution to this topic was published several years ago by Choudhary et al but has not been cited or discussed. If clinical application of the proposed methods has been used, please note and discuss it. |
Thank you for your comments.
We have included a reference to Choudhery et al in the first paragraph of the introduction and in the conclusion. |
Freezing and thawing of ADSC is simple and readily done as for any single cell solution. In fact, thousands of clinical applications of such cellular products are done successfully each year. Authors should note that no PI attempts cyropreservation without some CP added, so the comparisons on p14 are unrealistic. |
We have added the following to reflect your comments – ‘It is important to note that cryopreservation is not typically carried out without a CPA so there is limited utility in this comparison.’ |
A comparison of DMSO with internal versus external trehalose should be made. Discussion of different approaches on cell viability, proliferation and differentiation should be made.
|
The comparison was not done as there were too few studies available to draw any significant conclusions. Similarly, insufficient detail was given across the studies about cell proliferation and differentiation for comparison. |
Reviewer 3 Report
The systematic review entitled “Cryopreservation of human adipose tissues and adipose-derived stem cells with trehalose: A systematic review” by Conor Crowley, William PW Smith, Matthew KT Seah, Soo-Keat Lim and Wasim S Khan summarized the main articles about the potential benefits of Trehalose on Cryopreservation of Human Adipose Tissues and Adipose-Derived Stem Cells compared to DMSO.
While the review is well planned and results are clear, the authors can improve the manuscript with the following suggestions:
- Complete the introduction with the molecular characteristics defining mesenchymal stem cells such as cells surface markers, adhesion properties and immunomodulatory abilities.
- The authors could represent by a cartoon the mechanisms by which DMSO and trehalose act.
- Is there any evidence on the effect of Cryopreservation with trehalose with respect to differentiation towards osteoblats, chondrocytes or adipocytes? Does Cryopreservation of Human Adipose Tissues reduce the isolation efficiency of mesenchymal stem cells or their degree of purity? It would be very interesting if the authors discussed these topics.
- Authors could make tables clearer and more readable by using lines and colors.
- In the title, the authors should emphasize that the analysis covered by their work was conducted by comparing the freezing methods in DMSO vs trehalose.
The manuscript is well-written: English language and style are fine, minor spell check is required.
Author Response
REVIEWER 3
The systematic review entitled “Cryopreservation of human adipose tissues and adipose-derived stem cells with trehalose: A systematic review” by Conor Crowley, William PW Smith, Matthew KT Seah, Soo-Keat Lim and Wasim S Khan summarized the main articles about the potential benefits of Trehalose on Cryopreservation of Human Adipose Tissues and Adipose-Derived Stem Cells compared to DMSO. While the review is well planned and results are clear, the authors can improve the manuscript with the following suggestions:
|
Thank you for your comments.
We have included the following in the introduction ‘With regards to the identification of ADSCs, the Mesenchymal and Tissue Stem Cell Committee of the International Society for Cellular Therapy have proposed minimal criteria to define human mesenchymal stem cells. First, they must be plastic-adherent. Second, they must express the cell surface markers CD105, CD73 and CD90, and lack the expression of CD45, CD34, CD14 or CD11b, CD79alpha or CD19 and HLA-DR. Third, they must differentiate to osteoblasts, adipocytes and chondroblasts in vitro.’ |
|
Sugars have cryoprotective properties and are not toxic, hence trehalose has been investigated as an alternative CPA by virtue of its stability upon freezing. We have re-iterated in the text that the precise mechanism by which trehalose provides such protection is yet to be fully elucidated and have therefore not provided a graphic for this. |
|
We have included a brief reference to the effect of cryopreservation on the differentiation of MSCs in the discussion and highlighted that more work needs to be done in this area. ‘Finally, there remains some variation in cryopreservation of tissues and further studies are required to determine optimal protocols. Choudhery et al demonstrated that adipose tissue could be successfully cryopreserved without compromising cell morphology, as well as subsequent proliferation and differentiation potential, making this a useful tool in regenerative medicine. However, more work needs to be done on characterising the optimal cryopreservation techniques for the different tissues and cell types of interest.’ |
|
Thank you for the comment - the tables will be subsequently formatted to the Journal’s house style. |
In the title, the authors should emphasize that the analysis covered by their work was conducted by comparing the freezing methods in DMSO vs trehalose.
|
We have changed this to ‘Cryopreservation of human adipose tissues and adipose-derived stem cells with DMSO and/or trehalose: A systematic review’ |